# Food Accessibility and Nutritional Outcomes Among Food-Insecure Pregnant Women in Singapore

**DOI:** 10.3390/nu17050835

**Published:** 2025-02-27

**Authors:** Ethel Jie Kai Lim, Chengsi Ong, Nurul Syafiqah Said Abdul Rashid, Jeannette Jen-Mai Lee, Judith Chew, Mei Chien Chua

**Affiliations:** 1Department of Nutrition and Dietetics, KK Women’s and Children’s Hospital, Singapore 229899, Singapore; ong.chengsi@kkh.com.sg (C.O.); nurul.syafiqah.s.a.r@kkh.com.sg (N.S.S.A.R.); 2SingHealth Duke-NUS Maternal and Child Health Research Institute, KK Women’s and Children’s Hospital, Singapore 229899, Singapore; chua.mei.chien@singhealth.com.sg; 3Saw Swee Hock School of Public Health, National University of Singapore, Singapore 117549, Singapore; ephleej@nus.edu.sg; 4Department of Medical Social Work, KK Women’s and Children’s Hospital, Singapore 229899, Singapore; judith.chew.fh@kkh.com.sg; 5Department of Neonatology, KK Women’s and Children’s Hospital, Singapore 229899, Singapore; 6SingHealth Duke-NUS Paediatrics Academic Clinical Programme, Duke-NUS Medical School, Singapore 169857, Singapore

**Keywords:** food insecurity, pregnancy, micronutrient deficiency, nutrition

## Abstract

**Background/Objectives**: Food insecurity during pregnancy is associated with higher risks of negative physical outcomes for both mother and child. This study aims to understand experiences of food insecurity among low-income Singaporean pregnant women and its impact on nutritional status. **Methods**: In this cross-sectional, mixed-methods study, 49 food-insecure pregnant women were recruited from KK Women’s and Children’s Hospital between November 2021 and November 2023, among which 11 in-depth interviews were conducted. Questionnaires, anthropometric measurements, 24-Hour dietary recalls, metabolic and nutritional blood tests were conducted for all subjects. Descriptive quantitative analysis was performed and integrated with qualitative thematic analysis to explain findings. **Results**: On average, women were overweight pre-pregnancy (body mass index 26.1 ± 6.9 kg/m^2^) and had low haemoglobin and 25-hydroxyvitamin D levels. Calorie intake and intake from major food groups did not meet recommendations during pregnancy, except for “Grains”. From interviews, effects of financial constraints, how participants managed their food supply and pregnancy-related symptoms, supported findings from 24-Hour dietary recalls. **Conclusions**: Food insecurity led to suboptimal nutritional status and diets in Singaporean pregnant women despite appearing well-nourished. Further exploration of perspectives of food-insecure mothers, healthcare providers and welfare organisations is needed to devise long-term solutions to improve food security and alleviate malnutrition.

## 1. Introduction

The concept of the developmental origins of health and disease (DOHaD) proposes that environmental factors in periconception and early childhood can affect the risk of disease later in life [1]. One such factor closely associated with outcomes is nutrition, where under- or over-nutrition during conception and pregnancy have been associated with a greater risk of insulin resistance, obesity and hypertension in offspring. The International Federation of Gynaecology and Obstetrics (FIGO) recognises that micronutrient deficiencies can exist during pregnancy regardless of food security status [2]. Assessment of nutritional status is therefore recommended, with minimum screening for anaemia, so that appropriate interventions to enhance dietary diversity or supplementation can be provided. With limited access to nutritionally adequate and safe foods, food insecurity during pregnancy has been associated with higher risks of negative physical and psychosocial health outcomes for both mother and child [3]. For the mother, pregnancy complications include an increased risk of gestational diabetes mellitus (GDM), anaemia and pregnancy-induced hypertension. For the latter, they face an increased risk of birth defects, low birth weight and poorer developmental outcomes [3,4]. As an already physically and mentally challenging period, food insecurity can exacerbate stress and reduce a mother’s quality of life.

Within Singapore, food insecurity has been described as a condition in which an individual meets basic food intake requirements to curb hunger, is required to compromise on food quality such that cheaper options are inappropriate in sustaining health and/or makes decisions to sacrifice meal intake to prioritise other daily needs [5]. The Global Food Security Index, a measurement of affordability, availability, quality, safety and sustainability of a country’s food system, has consistently ranked Singapore’s affordability and availability of food highly [6], suggestive of low food insecurity rates. However, approximately 10% of households experience food insecurity and public awareness of the local situation is poor [7]. A report by the Lien Centre for Social Innovation identified low income to be a persistent contributor and that food-insecure Singaporeans are associated with having high body mass indices (BMIs), further masking awareness of food insecurity. This is consistent with the literature of both low- and high-income countries, yet the mechanisms of this incongruity between food insecurity and high BMI remain unclear [8]. Despite potentially poor health outcomes, lack of regulations around food assistance persists. There is no uniform criteria nor income threshold to determine eligibility of food assistance, and beneficiaries may receive a range of support at little to no cost, including frozen or shelf-stable meals, rations of non-perishable food or supermarket vouchers, and cooked meals that mainly target the elderly [5,9].

At present, most studies on factors associated with food insecurity in pregnancy among high-income countries have been concentrated within the United States and few within Asia [10]. There are a lack of data on the nutritional intake and health outcomes of food-insecure and low-income pregnant women in Singapore. Experiences around food insecurity and the appropriateness of food provisions from community food aid, if receiving any, are also unclear among this population. Given the extensive impact of DOHaD, the aim of this study was to better understand experiences of food insecurity among low-income Singaporean pregnant women and its impact on nutritional status during this critical period. Specific objectives are (i) to determine the health status and nutritional intake of these women and (ii) to explore experiences and pregnancy-related concerns, knowledge, and beliefs with food insecurity in this population.

## 2. Materials and Methods

### 2.1. Study Design and Recruitment Setting

This was a cross-sectional, convergent mixed-methods study. Quantitative and qualitative datasets were collected simultaneously but analysed separately. Findings from qualitative analyses were compared with quantitative findings where appropriate and assessed for complementarity or dissonance. The mixed-methods study design allows a better understanding of the complexities of experiences associated with food insecurity in pregnancy and their health-related consequences in this population of women.

Through convenient sampling, food-insecure pregnant women aged 50 years and below and between 20 and 40 weeks pregnant were recruited during outpatient appointments or hospitalisation at KK Women’s and Children’s Hospital (KKH), Singapore. Recruitment flyers were also electronically mailed to community service centres, women’s shelters and food charity organisations, which could refer clients for study recruitment. The Food Insecurity Experience Scale (FIES) developed by the Food and Agriculture Organisation (FAO) was used as a screening questionnaire, where at least one positive response to any of the questions was indicative of food insecurity [11]. All participants were reimbursed for their time and transport fares as appropriate. This study was reviewed and approved by the SingHealth Centralised Institutional Review Board (Reference Number: 2021/2291). Informed written consent was obtained from all participants, and all study procedures were conducted in accordance with the Helsinki declaration.

### 2.2. Quantitative Component

Upon recruitment, participants completed questionnaires, had anthropometrical assessments taken, and, where possible, had blood drawn for metabolic and nutritional tests.

The interviewer-guided questionnaire gathered demographic and pregnancy-related information, breastfeeding beliefs, as well as a 24-Hour dietary recall. Height was measured at the point of recruitment using the stadiometer available at clinics or at the ward, and reported weight before pregnancy was obtained. Blood pressure was measured using an upper arm blood pressure monitor (Omron HBP-1320).

Blood tests of interest were haemoglobin levels, 25-hydroxyvitamin D and fasting plasma glucose. However, blood tests were not conducted among participants who refused blood draws if they were formally recruited in their own homes or were opportunistically recruited at outpatient clinics. Fasting blood glucose was omitted if the participant was not fasted before blood draw. Normalcy of test results and its reference values were inferred from KKH’s Clinical Chemistry Laboratory, under the Department of Pathology and Laboratory Medicine.

#### Analysis of Quantitative Component

Nutritional intakes collected from the 24-Hour dietary recalls were analysed using FoodWorks v10 (Xyris Pty Ltd., Queensland, Australia) and compared with the Acceptable Macronutrient Distribution Ranges (AMDR) for macronutrients [12] and Singapore’s dietary guidelines for pregnant women [13]. Evidence synthesised by Mousa and colleagues [14] was also used to quantify recommended macronutrients during pregnancy. Micronutrients of interest include iron, zinc and calcium. Descriptive analyses were carried out for demographic information and nutrition-related parameters using Stata (Special Edition) version 15.1 (StataCorp LLC, Texas, USA). Categorical data were summarised as frequencies and percentages, while continuous data were summarised as means and standard deviations. Pre-pregnancy body mass index (BMI) was calculated based on the formula [weight (kg)/height (m^2^)] using reported weights and classified according to the World Health Organisation’s BMI ranges [15]. Associations between pre-pregnancy BMI categories and nutritional markers, blood pressure, energy intake and intake from each food group were assessed separately using one-way ANOVA and the Shapiro–Wilk test to check for normality. Statistical significance was defined with a *p*-value < 0.05.

### 2.3. Qualitative Component

Methods are reported according to the COREQ (COnsolidated criteria for REporting Qualitative research) Checklist [16] (Appendix A). Branching from epistemology, an interpretivist approach was used to understand nutrition-related beliefs and practices associated with food insecurity.

#### 2.3.1. Interview Design

A semi-structured interview guide was developed in accordance with clinical needs, as there was no locally available literature at the point of study conception, and experiences of food insecurity among pregnant women in high-income countries had not been well-studied. The goal of the interviews was to gain a deeper understanding of the participants’ social situations, including any food aid and financial assistance. Additionally, knowledge of nutritional and physical activity requirements during pregnancy, sources of pregnancy-related information, how the family navigates day-to-day food requirements, and any perspective on potential strategies to improve the nutritional statuses of pregnant women were ascertained. Pilot interviews were not conducted due to outreach limitations within the hospital and the culturally sensitive nature of food insecurity.

#### 2.3.2. Interview Team and Reflexivity

In-depth interviews were conducted by two female interviewers—JC (Master Medical Social Worker, Ph.D. in Social Work) and EJKL (Senior Clinical Dietitian, Master of Public Health). Where necessary, NSSAR facilitated Malay-to-English translation. At the start of interviews, participants were reassured that interviewers were not present in the capacity of their respective professions, and reasons for this study were reinforced. Clinical expertise in pregnancy-related nutrition concerns and social support systems available in Singapore allowed interviewers to obtain detailed information about food insecurity experiences.

#### 2.3.3. Data Collection

Interviews were conducted at the participants’ convenience and in person, either in private spaces at the hospital or participants’ homes. Spouses were allowed to be present if requested by the participant. Interviews lasted between one and two hours and were conducted largely in English, interspersed with Mandarin, Hokkien dialect, or Malay. Interviewees were free to ask questions and express concerns during the interview. Consent was sought to be audio-recorded, as was the option for interview cessation at any time. To ensure confidentiality, transcripts were anonymised, and information that participants requested to be removed was redacted.

#### 2.3.4. Analysis of Qualitative Component

Interviews were recorded in full and transcribed verbatim by EJKL. In view of the limited data available on food insecurity in Singapore, grounded theory techniques were adopted to allow themes to emerge from the data [17]. Interviews were coded line-by-line to identify pertinent themes relating food insecurity to nutrition and health in pregnancy, and an inductive approach was used for thematic analysis. Compilation and comparison of pertinent quotes relating to each subtheme were conducted using Microsoft Excel. Each excerpt in this paper includes the subject number and gestational age at the time of interview so that extracts can be linked to each participant while maintaining confidentiality.

## 3. Results

### 3.1. Participant Characteristics

General demographic characteristics of participants are summarised in Table 1. A total of 49 women were recruited within KKH between November 2021 and March 2024, while recruitment through electronic methods was unsuccessful. Participants were between 34 and 38 weeks of gestation at the point of recruitment and aged between 17 and 43 years old (29.8 *±* 5.9 years), with the majority being of Malay ethnicity (73.5%). Most women were unemployed (75.5%), and less than half the participants had completed post-secondary education (38.8%). All except one participant had an average monthly household income of no more than SGD 2000. Furthermore, 34.7% of participants had smoked tobacco or cigarettes or used nicotine patches during pregnancy. Among the eight women who consumed alcohol during pregnancy, five of them ceased further consumption after discovering their pregnancy status, two described themselves to be continuously drinking ‘heavy amounts’, while one had up to two drinks per week.

A total of 11 in-depth interviews were conducted between September 2022 and July 2023, and consisted of seven Malay, two Chinese and two of other ethnicities. Five of the women reported education up to secondary level, and most of the respondents have a household income of less than SGD 500 per month (*n* = 9). Only one interviewee was employed at the time of interview. Summarised characteristics of interviewees are included in Appendix A.

Table 2 provides an overview of pregnancy-related characteristics and measurements. The gestational age of first presentation to the obstetrician ranged between 2 and 26 weeks. Of the participants, 24.5% were first-time mothers, and four families had at least six children. The majority of the pregnancies were unplanned (79.6%). Apart from 10 women (20.4%) with known GDM, other health problems reported during pregnancy included hypertension, migraine, asthma and sexually transmitted infections. The majority of women reported consuming a pregnancy multivitamin (87.8%), with a lesser proportion taking other supplements such as folic acid, iron, fish oil, calcium, and vitamin D. The mean pre-pregnancy BMI of 26.1 kg/m^2^ (SD = 6.9 kg/m^2^) reflects that the women, on average, were overweight, with 48.9% of women being overweight or obese.

Blood tests were conducted for 41 participants and shown in Table 3. Women had low mean haemoglobin levels (10.7 ± 1.2 g/dL vs. laboratory reference 12.0–16.0 g/dL). Half the participants (50.0%) also had low levels of 25-hydroxyvitamin D (<20 ng/mL). No significant associations were found between pre-pregnancy BMIs and nutritional markers and blood pressure.

### 3.2. 24-Hour Dietary Recall

Nutrition-related quantitative results and their recommended intakes during pregnancy are summarised in Table 4. Mean calorie intake of women was 1748.5 ± 683.6 kcal, which is below the recommendation of 1800 kcal/day from the second trimester [13]. Percentage energy contributed by protein, carbohydrates and total fat intake were within the AMDR [12], and absolute protein intake of 71.6 g (SD = 31.7 g) met recommendations of 71 g/day during pregnancy [14]. Average contributions of energy from saturated fat and added sugar exceeded recommendations, but only approximately half the requirements for dietary fibre were met. None of the recommended servings for major food groups were met except “Grains”, which had an average intake of 7.5 ± 4.5 serves compared to the recommended six to seven per day. Sodium intake exceeded recommendations, while calcium, zinc and iron intakes did not meet recommendations during pregnancy. No significant associations were found between pre-pregnancy BMIs and energy intake, as well as intakes from each food group.

### 3.3. Qualitative Findings and Relationship with Quantitative Results Where Relevant

Three main themes emerged from the interviews: (i) coping strategies and food management, (ii) nutrition-related knowledge and its influences, and (iii) pregnancy-related symptoms affecting food intake.

#### 3.3.1. Theme 1: Coping Strategies and Food Management

Effects of financial constraints and how participants coped with managing their food supply supported quantitative results from the 24-Hour dietary recalls. Apart from financial constraints, personal motivations and preferences that affect food purchase, prioritisation of others’ needs, as well as perspectives on social support available are also described.

##### Subtheme 1a. Coping with Limited Resources

Low household income was the primary reason for food insecurity, as the ability to purchase food was dependent on what funds were left after other household expenses were accounted for. This was also consistent with quantitative data on reported household income and participants’ employment statuses. To save costs, participants would compare product prices between and within supermarkets, opting for frozen instead of fresh items:


*“See which one got discount then I buy discount thing. …You see the two-for-how-much-how-much, I buy two, sometimes buy four.”*

*(PW014, 39 weeks)*



*“Currently I don’t have much meat because meat is expensive.”*

*(PW024, 36 weeks)*


Red meat was avoided due to its higher cost compared to other proteins, and its longer cooking time meant an increase in utility bills. Careful planning of meals to provide equal portions to each family member served to reduce food wastage but also meant that individuals may not always be satiated (see Appendix A). Furthermore, most participants would consume the same food throughout the day or skip meals to avoid purchasing additional ingredients:


*“Like currently, usually, we usually eat like easily…at most once, once a day. Usual one lah, it’s usually once a day.”*

*(PW032, 36 weeks)*


Given the unpredictability of having sufficient food, participants preferred to stock up on items with longer shelf lives. Dry and frozen food would therefore take precedence over fresh fruits and vegetables, given its perishability. This is reflected in the 24-Hour dietary recall, where participants met recommended intakes of carbohydrates and grains, while intake of fruits and vegetables, including dietary fibre, fell below recommendations. In addition, the reported low intake of red meat, fruit, and vegetables was consistent with zinc and iron intake not meeting recommended amounts during pregnancy.

##### Subtheme 1b. Personal Motivations and Preferences

Ease of meal preparation was a consideration for several participants, especially if traditional dishes required multiple ingredients, which would incur additional cost and effort:


*“Steamed stuff is like, you just put inside the fish then you steam then everything is done. This one must wait for the oil to come out lah, cannot chao tah (burn), cannot this, cannot that. Then I’m like, huh! Must blend this, must blend like 101 ingredients.”*

*(PW024, 36 weeks)*


The use of instant pastes and pre-made soups helped to expand the variety of meals that can be prepared with minimal ingredients. In instances where participants were picky with vegetables at baseline, the decision to prepare vegetables for the day was mood-dependent:


*“Vegetables, yeah, also depends on the day. On the day, If I say I want to cook vegetable, I cook. If not, I don’t.”*

*(PW036, 35 weeks)*


Apart from financial concerns, such food preferences further support why the intake of vegetables was inadequate. On the other hand, some participants who did not enjoy vegetables pre-pregnancy had become more health conscious and recognised the importance of nutrition with age and grand multiparity:


*“Because the age is getting higher then this one also different from the previous six, so the diet, the healthy lifestyle is all different. It’s totally different, you know? …I never eat the vegetable, suddenly eat all this. Ok lah ok lah, much more better. At least healthy healthy lifestyle. At least got the vegetable, got the fruits.”*

*(PW032, 36 weeks)*


##### Subtheme 1c. Prioritising Others’ Needs over Own

Particularly among those who already have children, participants placed themselves as less of a priority and were willing to compromise on personal nutrition to ensure that their children had adequate food. As explained by one of the interviewees:


*“Always let my kids eat more…If I hungry, I will think of…see if my house got bread. (In Mandarin) My mother always said to let children eat first…to only care about myself? I don’t know lah, cannot do this kind of thing.”*

*(PW014, 39 weeks)*


Furthermore, children’s preferences often influenced what was prepared at home, which was recognised by participants to have been in place of more healthful options:


*“So there is times where I don’t buy veggie at all. So I just buy something that the kids will eat together. Those kind fishball, fishcake, hotdog? Yeah, all those unhealthy.”*

*(PW026, 39 weeks)*


For some participants, the cost of maternal milk was a limitation despite knowledge of its nutritional benefits. Formula milk for children would instead be prioritised regardless of its necessity:


*“And you know especially they say drink the pregnancy milk, you know or not, one tin, I can buy one formula milk for my children. Might as well I give my children than I drink myself.”*

*(PW029, 35 weeks)*


Where they were able to source free samples of maternal milk, it was recognised that supply would not suffice throughout the pregnancy, and consumption of maternal milk was helpful in an attempt to ‘at least fill up their tummy’ in instances where meals had to be skipped.

##### Subtheme 1d. Social Support and Food Rations

Spouses, extended family and friends were primary sources of support. Among most families, husbands were either sole breadwinners or would assist with grocery shopping and meal preparation. If necessary, parents or in-laws may be a source of financial assistance or as food providers. As expressed by participants:


*(Referring to her father): “I don’t need tell him how much I want, he just gives. SGD 100, 200, he’ll give.”*

*(PW036, 35 weeks)*



*We will find other alternative, like if say cannot borrow right, will ask his mom cook.”*

*(PW013, 36 weeks)*


For some, workplaces that involve handling food sometimes had excess supply or foods of short expiry that employees were able to take home. Such instances would provide relief to families, albeit limited control over the nutritional quality of such items:


*“Sometimes you know how they work groceries like, got dent a little bit in the vegetable also throw right. So he will bring back lah. So that’s where we usually get our vegetable source lah.”*

*(PW029, 35 weeks)*



*“Yeah, my husband brings back some of ingredients from work. The frozen food, the ones you can just fry. The…what? Potato wedges, satay sticks…satay…Yeah, he took from his supplier. That one also sometimes the kids eat, so we will share lah.”*

*(PW033, 37 weeks)*


Welfare organisations were a common source of food rations for participants. However, engagement with welfare organisations drew mixed responses. For some, social workers from Family Service Centres (FSCs) would assist with providing the family with dry food rations. These include rice, instant noodles, cooking oil, canned food and biscuits. While participants expressed gratitude for the assistance, rations were perceived as insufficient, misaligned with food preferences, or culturally inappropriate at times:


*“Sometimes they got give the thing that I don’t eat lah. Or the people around the house don’t eat. …Peanut with the sauce that one. Don’t know how to read the label.”*

*(PW008, 38 weeks)*



*“So she (social worker) send, and I’m very grateful that I had, I had some extras. But I look at it, I was like…but for the whole family, one day…one meal we all just take the Maggi mee finish already. (laughs)”*

*(PW015, 36 weeks)*


Only one participant had ever received fresh rations, while another received supermarket vouchers once from a church. Conversely, the few participants with limited family support and who were single parents were unable to obtain financial assistance due to their employment status. Also, some participants were aware that they would not fulfil criteria for assistance or were adamant about disengaging with any form of assistance due to stigma experienced.

In identifying beneficial food rations, most participants nevertheless preferred dry rations or frozen meats due to their longer shelf lives, compared to fresh rations. Some participants also cited the availability of breakfast items such as bread, breakfast spreads, and eggs as a contributor to consistent breakfast intake. Assistance in the form of supermarket vouchers was considered helpful by some, especially in providing autonomy over food choices when faced with necessitous circumstances:


*“Like NTUC, Giant, this kind of thing. At least I can go and purchase healthy food for myself…So at least like those vouchers that they give, I can stock up food for me to eat during my confinement period when I’m unable to go to work.”*

*(PW016, 34 weeks)*


#### 3.3.2. Theme 2: Nutrition-Related Knowledge and Its Influences

This theme summarises participants’ perceptions and knowledge of healthy eating, their trusted sources of information and preferred methods of receiving information.

##### Subtheme 2a. Perceptions of Healthy Food

Inclusion of fruits and vegetables was most cited as what a healthy diet should comprise. Among those who expressed pickiness around vegetables, fruits were viewed as a suitable replacement:


*“Because I don’t eat vegetables, so I try to eat fruits. But also, not everyday…Not a habit.”*

*(PW033, 37 weeks)*


Commonly mentioned cooking methods that were perceived to be healthy were those that used little oil, salt and sugar. Participants who were adept at cooking would make use of aromatics and spices to enhance the flavour of dishes to reduce the amount of salt used in meal preparation:


*“Mostly I cook soup, because I very lazy. …Yeah, everything just throw inside. So I just use all the fish ok, tauhu ok, broccoli ok, a bit more healthier one lah…then don’t use oil, like that. Boil with garlic and you know, ginger for the flavour, like that.”*

*(PW015, 36 weeks)*


In addition to food types, portion and consistency of mealtimes were deemed important to some. While they acknowledged some understanding of healthy eating principles, there was an inability to translate knowledge to practice, as satiety of the children was more important:


*“Usually, what I know for nutrition is like, the way you eat have to be balanced…How you consume the food, the portion, yeah…in the sense that for like, normal rice, instead of normal rice, take brown rice…Veggie? And must also have some meats. And also wise in drinks, have to be plain water, yeah. Or milk. Yeah. Fruits? (Laughs) But we don’t have that…Don’t have proper meal at all, just chinchai chinchai (Singlish meaning ‘whatever’) eat like that lah. As long as it’s err…full for my kids lah.”*

*(PW026, 39 weeks)*


Some women also reported limiting sugar intake in fear of developing diabetes. In contrast, individuals who consumed sugar-sweetened beverages justified their intake with being physically active or using it to tide through a skipped meal.

##### Subtheme 2b. Sources of Information

Family and friends were key sources of information for several participants, particularly if there were adverse outcomes to learn from:


*“My sister go and eat raw egg because she don’t know. She go and eat raw egg then she miscarriage.”*

*(PW008, 38 weeks)*


For others, medical conditions such as high blood pressure and diabetes within the family provided the chance to receive nutrition-related information from health professionals, therefore contributing to changes in eating habits at home. The obstetrician was also frequently cited as a trusted professional and source of reassurance, as described:


*“As long as you go every time got check-up, you go check-up. Doctor say baby ok, you ok. So whatever you’re eating all this is still ok.”*

*(PW029, 35 weeks)*


There was an understanding that their nutritional intakes were suboptimal, and doctors’ visits allowed them to obtain pregnancy multivitamins. This was demonstrated in the high compliance with the supplement (90.2%) and mean intakes of micronutrients from food alone being less than recommended during pregnancy:


*“For pregnancy, the doctor got give (supplements), I got eat. For the calcium one, I also got take.”*

*(PW014, 39 weeks)*


However, only one interviewee cited KKH as a source of nutrition-related information. The most pervasive source was digital media, which ranged from televised programmes, ‘Google’ searches and social media platforms that either provided information or allowed participants to connect with other women and share pregnancy- and breastfeeding-related information. Although participants acknowledged that information online was sometimes inconsistent and may not have been credible, it was nonetheless a convenient source:


*“I think that because people go KK is to see doctor, it’s not really to go and share our story to them. So I think that’s why they would rather find the closest friends, or Google lor.”*

*(Husband of PW033, 37 weeks)*


Among the women interviewed, some had used mobile applications to track their pregnancies and felt that it would be useful for KKH to develop a comprehensive application that not only includes recommendations for a healthier lifestyle during pregnancy but also expectations during each trimester and how to care for newborns. However, there were different preferences in the way information is received, and one-to-one consultations could be helpful in motivating the adoption of healthier lifestyles, as explained:


*“I prefer one-to-one because can explain. Because I don’t really understand when I read…And one thing good about the one-to-one right, they also can ask you—how is it going with your this and that? They want to know how is it going, and you know the feeling, like, you know…the feeling…you feel more and more… want to do this.”*

*(PW013, 36 weeks)*


#### 3.3.3. Theme 3: Pregnancy-Related Symptoms Affecting Food Intake

This theme describes changes in food intake contributed by symptoms experienced during pregnancy. Several participants described taste changes and having specific cravings during pregnancy, of which purchases had to be limited due to financial constraints, potentially leading to feelings of helplessness and frustration in trying to satisfy cravings:


*“You know, I want to eat something lighter, not all this rice all so heavy this kind of thing. But by the time comes, I’m eating something heavy because of this craving that comes in. If it’s not ayam penyet, it’s mutton penyet. I say this is crazy.”*

*(PW015, 36 weeks)*



*“Because I didn’t really have enough money and stuff like that. So initially it was a bit difficult lah. So I only crave for those things that I could afford, and then those things when I don’t get, I also get frustrated.”*

*(PW016, 34 weeks)*


Mostly in the first trimester, issues with morning sickness and tolerance to specific foods also limited the variety and quantity that could be consumed. For some, despite the recognition of nutritive food items required during pregnancy, cravings tended to be energy-dense, nutrient-poor options that may replace necessary food groups:


*“Then now, I drink more sweet drinks, I get worried also lah. Because I drink less milk.”*

*(PW013, 36 weeks)*



*“Because pregnancy also, this one don’t like milk ah. I also don’t know, this one likes bandung ah…if money have, I buy. If not, then don’t have lah.”*

*(PW036, 35 weeks)*


This was also reflected in analysis of added sugars comprising a mean of 11.8% (SD = 10.0%) of energy intake, which exceeded recommendations for daily sugar limit. Although milk was not tolerated well by several participants, there was little mention of calcium-rich alternatives.

## 4. Discussion

This study is the first to provide insights into food insecurity and its effect on nutritional outcomes among pregnant women in Singapore. With majority of women in the healthy to overweight and obese BMI range pre-pregnancy, issues with food security are easily masked by the appearance of being well-nourished. Strategic decisions to select cheaper food with long shelf lives and prioritisation of satiety often led to relinquishment of nutrient-dense, but perishable, fruits and vegetables, albeit with recognition that these foods contribute to healthy eating practices. Despite high compliance with pregnancy multivitamins, the nutrient composition of supplements may still be inadequate to prevent micronutrient imbalances within this population of women, as seen in the low haemoglobin and 25-hydroxyvitamin D levels. Pregnancy-related symptoms, such as morning sickness and taste changes, further inhibit intakes of nutrient-dense, balanced meals.

In other high-income countries, a meta-analysis by Nguyen and colleagues found significant associations between food insecurity and pre-pregnancy obesity, but not overweight or underweight BMI [19]. In studies where pre-pregnancy BMI was reported as a continuous measure, BMI was significantly higher among food-insecure women compared with food-secure women. In this study, the majority of women were not underweight, which aligns with literature that food insecurity in a high-income country is not associated with low weight status. Pertaining to GDM as an adverse pregnancy outcome, a prevalence of 20.4% in this study is slightly higher than the population norm of 18.9% [20]. With prudent dietary patterns having been associated with lower risk of developing GDM [21], the poor adherence to dietary guidelines in pregnancy among study participants could have placed them at a higher risk of GDM. Despite recommendations for universal screening [18], 20.4% of women also reported being unaware of their GDM status. Nevertheless, the association between food insecurity and GDM status remains inconsistent [22,23]. Haemoglobin levels were of a lower concentration among 85.4% of women in this study. Among general adults and pregnant women, lower concentrations of haemoglobin and iron-deficiency anaemia have been associated with food insecurity [24,25]. However, as iron status has been found to decline with pregnancy progression among Singaporean women [26], the ability to characterise low haemoglobin levels as a marker of food insecurity remains in question, given that women were largely recruited near term. Furthermore, 50.0% of participants were vitamin D deficient, which is higher than the 13.2% found in a local study among women of 26 to 28 weeks of gestation [27]. Given that vitamin D insufficiency and deficiency were more common among Malay and Indian women, the high percentage of participants being Malay in this study could explain the higher prevalence of vitamin D deficiency. In relation to dietary outcomes, a consistent pattern of low fruit and vegetable intake among food-insecure pregnant women has been found [18]. Red and processed meat intake were also significantly higher among some food-insecure groups compared with food-secure groups. However, as studies of food insecurity in high-income countries were primarily conducted in the West, applicability of findings to the Singapore context should be interpreted with caution due to cultural differences.

Qualitative findings were consistent with literature—low income was the leading factor associated with food insecurity during pregnancy, and strategies such as skipping meals or reducing the number of meals per day, purchasing cheaper food, and turning to family and friends for support were coping mechanisms to ensure satiety, especially in prioritising the needs of children above self [10,28]. Similarly, desire for optimal infant outcomes was a motivation to eat healthily, yet pregnancy symptoms posed as a barrier to meal preparation, together with affordability of healthier food items. The difference is that breakfast was frequently missed among participants but was seen as important for pregnancy health in other studies [29]. Among participants who sought food assistance, rations were perceived as sporadic, inadequate, and misaligned with needs, which was also common feedback identified in The Hunger Report [7]. While the main reason for not seeking food support in this study is unclear, ‘embarrassment’ and lack of awareness of available food support were main reasons among the wider Singapore population. Although food rations are a stopgap measure that provide temporary relief to mothers, rations provided should be culturally appropriate, nutritious, and aligned with preferences. An unexpected finding from the interviews was the lack of resources readily available in KKH, which resulted in the majority of participants sourcing for information online that could contain both misinformation and disinformation. Addressing misinformation and linking these women to reputable sources of information are key to building knowledge and driving positive lifestyle behaviours in pregnancy. Additionally, families that have adopted successful strategies to achieve better nutrition in pregnancy can serve as guidance to other families that may struggle with prioritising nutrition and meal planning. While some participants had proposed ideas of what resources may be beneficial during their pregnancy journey, further needs analysis is required.

The core strength of this study is its mixed-methods design, which provided an understanding of how food supply and nutrition are maximised by food-insecure pregnant women in the face of financial adversity. Qualitative data explained the constraints and considerations in making food choices, which complemented quantitative findings of suboptimal intakes and nutritional statuses. Identification of barriers to healthy eating is required to understand the needs of this population of women so that gaps can be addressed through targeted interventions. Other than food rations, provision of shopping vouchers, vouchers specific to purchase of fruits and vegetables, and commercial meal kits are examples of alternative food support that can be explored [30,31,32]. To determine areas for improvement in healthcare, further research is required to explore awareness and roles of healthcare providers in identification and management of food-insecure women antenatally. On a macro level, nutrition guidelines for food assistance programmes or designated food packages can be considered to improve the quality of rations received by pregnant women, which have been shown to improve diet quality and birth outcomes [33].

Not without limitations, the small sample of this study likely resulted in wide distributions of parameters analysed, as well as the lack of power to detect any significant associations between pre-pregnancy weight status and nutritional outcomes. It was also not meaningful to classify the severity of food insecurity with responses to the FIES given the small sample size. Potentially, as a result of stigma and embarrassment faced by food-insecure individuals, recruitment of participants was challenging and constrained to KKH. It is acknowledged that social desirability bias could have led to over-reporting compliance with nutritional supplements, despite anonymity emphasised throughout data collection. While this study has attempted to characterise the nutritional intakes and statuses of these women, it is acknowledged that 24-Hour dietary recalls do not account for day-to-day variation in intakes, and other biomarkers should be explored in future studies. From the interviews, data saturation was unlikely to be reached due to the small sample size. Given that recruitment was already challenging, purposive sampling of interviewees by family structure would be proven more difficult.

## 5. Conclusions

The current study demonstrates food choices and limitations owing to financial constraints and pregnancy-related symptoms, thereby contributing to imbalanced diets that result in suboptimal nutritional status in pregnancy. With light shed on the complexities of food insecurity among pregnant women within the Singapore context, interventions need to be tailored to address their unique needs in support of optimal birth outcomes. In considering the utility of rations in alleviating food insecurity, food items should ideally be matched to the family’s needs, especially in managing pregnancy-related taste changes. Collaboration between mothers, healthcare providers and welfare organisations is needed to devise long-term solutions that improve food security and alleviate malnutrition.

## Figures and Tables

**Table 1 nutrients-17-00835-t001:** Demographic profiles of participants (n = 49).

Characteristics	Mean ± SD or n (%)
Age at the time of interview (years)	29.8 ± 5.9
Ethnicity	
Chinese	4 (8.2)
Malay	36 (73.5)
Indian	4 (8.2)
Others	5 (10.2)
Highest education level	
Primary	5 (10.2)
Secondary	25 (51.0)
Post-Secondary	19 (38.8)
Employment status	
Unemployed	37 (75.5)
Employed	12 (24.5)
Average household income (SGD) per month	
Less than SGD 500	28 (57.1)
SGD 500–1000	13 (26.5)
More than SGD 1000	7 (14.3)
Not reported	1 (2.0)
Smoking status during pregnancy	
Smoker	17 (34.7)
Non-smoker	32 (65.3)
Ever consumed alcohol during pregnancy	
No	41 (83.7)
Yes	8 (16.3)

**Table 2 nutrients-17-00835-t002:** Gestational characteristics and anthropometric measurements of participants (n = 49).

Characteristic	Mean ± SD or n (%)
Gestational age at the time of interview (weeks)	37.0 ± 1.5
Gestational age at the first doctor’s visit (weeks)	9.7 ± 6.0
Number of children	
0	12 (24.5)
1	8 (16.3)
2	13 (26.5)
≥3	16 (32.7)
Planned pregnancy	
No	39 (79.6)
Yes	10 (20.4)
Diagnosed with gestational diabetes mellitus (GDM) ^#^	
No	29 (59.2)
Yes	10 (20.4)
Not reported/known	10 (20.4)
Diagnosed with other health problems	
No	34 (69.4)
Yes	15 (30.6)
Supplement intake during pregnancy	
Folic acid	36 (73.5)
Iron	18 (36.7)
Calcium	7 (14.3)
Vitamin D	8 (16.3)
Multivitamin	43 (87.8)
Fish oil	25 (51.0)
Pre-pregnancy body mass index (BMI) (kg/m^2^) based on the reported weight	26.1 ± 6.9
<18.5	5 (10.2)
18.5–24.9	18 (36.7)
25.0–29.9	11 (22.4)
≥30.0	13 (26.5)
Not reported/known	2 (4.1)
Blood pressure (mmHg)	
Systolic	110.7 ± 13.2
Diastolic	69.0 ± 9.6

^#^ GDM is diagnosed from 24 weeks of pregnancy based on the International Association of Diabetes in Pregnancy Study Group’s criteria where one or more of the following criteria are met: fasting plasma glucose of 5.1–6.9 mmol/L; 1-hour plasma glucose of 10.0 mmol/L following a 75-gramme oral glucose load; 2-hour plasma glucose of 8.5–11.0 mmol/L following a 75-gramme oral glucose load [18].

**Table 3 nutrients-17-00835-t003:** Biomarker profiles of participants.

Biomarkers	n	Normal Readings (%)	Mean ± SD	Reference Values
Haemoglobin (g/dL)	41	14.6	10.7 ± 1.2	12.0–16.0
25-Hydroxyvitamin D (ng/mL)	38	50.0	19.2 ± 8.4	20.0–100.0
Fasting plasma glucose (mmol/L)	30	96.7	4.2 ± 0.6	≤5.0 *

* Reference value specific to pregnancy.

**Table 4 nutrients-17-00835-t004:** Mean nutrient intake with reference to daily recommendations during pregnancy (n = 49).

24-Hour Recall Data	Mean ± SD	Recommendations
Energy intake (kcal)	1748.5 ± 683.6	1800–2250 kcal/day ^a^
Macronutrients		
Protein intake (g)	71.6 ± 31.7	71 g/day ^b^
% kcal from protein	16.5 ± 4.7	10–35% ^c^
Total fat (g)	63.1 ± 33.0	N.A.
% kcal from total fat	31.1 ± 8.8	20–35% ^c^
Saturated fat (g)	21.3 ± 11.1	N.A.
% kcal from saturated fat	10.6 ± 4.2	<10% ^c^
Carbohydrates (g)	219.1 ± 89.6	175 g/day ^b^
% kcal from carbohydrates	51.2 ± 9.4	45–65% ^c^
Dietary fibre (g)	15.2 ± 8.1	28 g/day ^b^
Added sugar (g)	52.1 ± 44.4	N.A.
% kcal from sugar	11.8 ± 10.0	<10% ^c^
Servings of major food groups		
Grains	7.5 ± 4.5	6–7 ^a^
Fruit	0.4 ± 0.7	2 ^a^
Vegetables	1.9 ± 2.1	3 ^a^
Protein	1.8 ± 1.2	2.5 ^a^
Dairy products	0.4 ± 0.5	1 ^a^
Micronutrients (minerals) (mg)		
Calcium	645.7 ± 417.1	1000 ^a^
Zinc	7.8 ± 3.5	10 ^a^
Iron	10.9 ± 7.4	27 ^a^
Sodium	2410.5 ± 1487.3	2000 ^a^

^a^ Nutritional recommendations during pregnancy available from the Ministry of Health, Singapore (HealthHub) [13]. ^b^ Recommended quantity of macronutrients during pregnancy [14]. ^c^ Recommendations based on the acceptable macronutrient distribution range [12].

## Data Availability

Datasets will be available upon reasonable request from the authors.

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
