# Peer review of "Food Accessibility and Nutritional Outcomes Among Food-Insecure Pregnant Women in Singapore"

_nutrients, 2025, doi:10.3390/nu17050835_

Round 1
Reviewer 1 Report
Comments and Suggestions for Authors
This paper addressed important issues related to food accessibility and nutritional outcomes for pregnant women, as well as the high relevance of the Nutrients journal to the readers. The paper is well writing with sound methodology and analyses. There are a few recommendations to be offered to increase overall clarity of this important work. First, the abstract is quite long and should be substantially reduced. It also does not need to have the section headings in the abstract (e.g., results, methods, etc.).
Within the introduction, the reference for the Food Security Index is provided but some additional background on why this relative to other well-known measures would strengthen the introduction. The authors mention that there is a lack of data on nutritional intake for women in Singapore. Additional information as to whether this is unique to this country or is relevant to other nations with similar characteristics (and what those dimensions are).
Within the methods section, additional details on the nature and process of the "convergent" mixed methods approach would be helpful since this is not a traditional and/or common methodology. Additional detail on why and how "grounded theory techniques were adopted" so again, there is clarity and detail to help readers understand the selection of approaches to data collection and methods used. The section on data analysis is very brief, and additional details on the specifics of both qualitative and quantitative analyses are needed.
The data tables are well organized however some additional footnotes would be helpful especially for those outside of the medical field. The themes that were discovered and summarized in the paper are quite strong and very helpful for the reader to understand the range and depth of the information collected from participants.
However, this clarity is lacking within the discussion section of the paper. It appears not to follow or capture those key themes from the results section and discuss their implications for future research. One suggestion may be to highlight some macro-level themes related to the research findings to help organize and enhance the clarity of the discussion section. For example, the discussion could focus on resources, access, personal needs, and obstacles as one example.
Author Response
Thank you for taking the time to review this manuscript. Please find the responses below and the corresponding revisions in the re-submitted file. Main changes have been highlighted and the changed tracked in the re-submitted file.
Comment 1: This paper addressed important issues related to food accessibility and nutritional outcomes for pregnant women, as well as the high relevance of the Nutrients journal to the readers. The paper is well writing with sound methodology and analyses. There are a few recommendations to be offered to increase overall clarity of this important work. First, the abstract is quite long and should be substantially reduced. It also does not need to have the section headings in the abstract (e.g., results, methods, etc.).
Reply 1: We have removed section headings and shortened the abstract. Details of nutritional intakes were summarised and qualitative results within the abstract were summarised.
Comment 2: Within the introduction, the reference for the Food Security Index is provided but some additional background on why this relative to other well-known measures would strengthen the introduction.
Reply 2: Thank you for pointing this out. The Global Food Security Index (GFSI) focuses on affordability, availability, quality and safety and sustainability of food systems, and is a commonly used indicator of a country’s food system. This has been added into the revised manuscript to show why the GFSI was used to suggest low food insecurity rates. This revision can be found in page 2, line 58: “The Global Food Security Index, a measurement of affordability, availability, quality, safety and sustainability of a country’s food system, has consistently ranked Singapore’s affordability and availability of food highly [6], and suggestive of low food insecurity rates.”
Comment 3: The authors mention that there is a lack of data on nutritional intake for women in Singapore. Additional information as to whether this is unique to this country or is relevant to other nations with similar characteristics (and what those dimensions are).
Reply 3: Agree that this could have been elicited more clearly. We have added a statement to highlight a lack of studies in high-income countries, especially within the region. This revision can be found in page 2, line 73:
“At present, most studies on factors associated with food insecurity in pregnancy among high-income countries have been concentrated within the United States, and few within Asia [10]. There is a lack of data on the nutritional intake and health outcomes of food insecure and low-income pregnant women in Singapore.”
Comment 4: Within the methods section, additional details on the nature and process of the "convergent" mixed methods approach would be helpful since this is not a traditional and/or common methodology.
Reply 4: We have removed section 2.4 (Mixed methods reporting) and merged it with 2.1 (Study design) to consolidate the nature and process of the convergent mixed-methods approach. We have specified that the data sets were collected simultaneously, and elaborated on why the mixed-methods design was chosen. The change can be found in page 2, line 86:
“Quantitative and qualitative data sets were collected simultaneously but analysed separately. Findings from qualitative analyses were compared with quantitative findings where appropriate, and assessed for complementarity or dissonance. The mixed-methods study design allows a better understanding of the complexities of experiences associated with food insecurity in pregnancy and their health-related consequences in this population of women.”
Comment 5: Additional detail on why and how "grounded theory techniques were adopted" so again, there is clarity and detail to help readers understand the selection of approaches to data collection and methods used.
Reply 5: Grounded theory technique was used as there is currently little data on food insecurity in Singapore and analysis was primarily inductive, which is a key feature of grounded theory. We have revised the manuscript to reflect that the technique was adopted to allow themes to emerge from the data. This can be found in page 4, line 170:
“In view of the limited data available on food insecurity in Singapore, grounded theory techniques were adopted to allow themes to emerge from the data (17). Interviews were coded line-by-line to identify pertinent themes relating food insecurity to nutrition and health in pregnancy, and an inductive approach was used for thematic analysis.”
Comment 6: The section on data analysis is very brief, and additional details on the specifics of both qualitative and quantitative analyses are needed.
Reply 6: Section 2.2.1 and 2.3.4 describe quantitative and qualitative analyses respectively. Use of the Shapiro-Wilk test has been added into the quantitative component (page 3, line 133), and we have elaborated on the use of grounded theory techniques (Reply 5; page 4, line 170).
Comment 7: The data tables are well organized however some additional footnotes would be helpful especially for those outside of the medical field.
Reply 7: We noted that the '≥' sign in Table 2 were reflected as ‘3’ and has been amended. An explanation of the diagnostic criteria for GDM has been added as a footnote in Table 2 (page 6).
Comment 8: The themes that were discovered and summarized in the paper are quite strong and very helpful for the reader to understand the range and depth of the information collected from participants. However, this clarity is lacking within the discussion section of the paper. It appears not to follow or capture those key themes from the results section and discuss their implications for future research. One suggestion may be to highlight some macro-level themes related to the research findings to help organize and enhance the clarity of the discussion section. For example, the discussion could focus on resources, access, personal needs, and obstacles as one example.
Reply 8: Thank you for highlighting this. We have included:
(i) A statement to point out the need to be culturally-appropriate and sensitive to pregnancy-related changes if providing food rations (page 14, line 493):
“Although food rations are a stopgap measure that provide temporary relief to mothers, rations provided should be culturally appropriate, nutritious, and aligned with preferences.”
This statement was initially positioned in the conclusion but we have summarised it in the conclusion instead (page 15, line 539):
“In considering the utility of rations in alleviating food insecurity, food items should ideally be matched to the family’s needs, especially in managing pregnancy-related taste changes.”
(ii) Suggestions to address coping strategies and nutrition-related knowledge deficit in this population, which relate to themes 1 and 2 on ‘Coping strategies and food management’ and ‘Nutrition-related knowledge and its influences’ respectively. This revision can be found in page 14, line 498:
“Addressing misinformation and linking them to reputable sources of information are key to building knowledge and driving positive lifestyle behaviours in pregnancy. Additionally, families that have adopted successful strategies to achieve better nutrition in pregnancy can serve as guidance to other families that may struggle with prioritising nutrition and meal planning.” Transition words in subsequent sentences were also revised to improve the flow of the manuscript.
Reviewer 2 Report
Comments and Suggestions for Authors
It is a well written MS which interpret the attributable risk factors and characteristics of insecure food intake during pregnancy in Singapore.
I would like to suggest the MS for publication because it is a scarcely investigated area and the authors describe some specifics in this field. I have just some minor remarks.
1) The authors state that they used ANOVA as statistical methods. Were the normality checked?
Some typographic errors:
2) line 171 gestation age --> gestational age
3) line 226 "Grains"
It seems that the pregnant women with GDM did not comply with the diet restriction or the motion requirements.
The authors should speculate around that the unhealthy food might increase the CH-intake put pregnant women at a higher risk of GDM.
As I understand, the authors determined the iron serum level during pregnancy and classified as a nutritional status indicator. Low Hb is marked as a characteristic of a food insecure habit (line 470). As I know , the overwhelming majority of pregnant women is at a risk of low Hb and iron deficiency with adequate iron intake in most of the western countries. So it is questionable how the low iron and Hb level can be pertained to food insecurity.
Author Response
Thank you for taking the time to review this manuscript. Please find the responses below and the corresponding revisions in the re-submitted file (highlighted). Main changes have been highlighted and the changed tracked in the re-submitted file.
Comment 1: The authors state that they used ANOVA as statistical methods. Were the normality checked?
Response 1: Normality was checked. The manuscript has been revised to include the use of Shapiro-Wilk test to check for normality. This can be found in page 3, line 133:
“Associations between pre-pregnancy BMI categories and nutritional markers, blood pressure, energy intake and intake from each food group were assessed separately using one-way ANOVA, and the Shapiro-Wilk test to check for normality.”
Comment 2: typographic error: (1) line 171 gestation age -> gestational age; (2) line 226 "Grains"
Response 2: Thank you for pointing these out. They have been amended in:
(1) Page 4, line 175: “Each excerpt in this paper includes the subject number and gestational age at point of interview…”
(2) Page 7, line 223: “None of the recommended servings for major food groups were met except “Grains”, which had an average intake…”
Comment 3: The authors should speculate around that the unhealthy food might increase the CH-intake put pregnant women at a higher risk of GDM.
Response 3: Agree. We have made the revision to include how poor adherence to dietary guidelines could have placed them at a higher risk of GDM. A reference to a systematic review on the maternal diet in the development of GDM has been included. This can be found in page 13, line 517:
“With prudent dietary patterns having been associated with lower risk of developing GDM [20], the poor adherence to dietary guidelines in pregnancy among study participants could have placed them at a higher risk of GDM.”
Comment 4: As I understand, the authors determined the iron serum level during pregnancy and classified as a nutritional status indicator. Low Hb is marked as a characteristic of a food insecure habit (line 470). As I know , the overwhelming majority of pregnant women is at a risk of low Hb and iron deficiency with adequate iron intake in most of the western countries. So it is questionable how the low iron and Hb level can be pertained to food insecurity.
Response 4: Thank you for highlighting this. In Page 13, line 463, “While iron levels were normal” was removed from the sentence as iron levels were not described in the results. We agree that iron deficiency and low haemoglobin levels are common in pregnancy and difficult to make a direct link to food insecurity. This has been suggested in the revision in page 13, line 466:
“However, as iron status has been found to decline with pregnancy progression among Singaporean women [26], the ability to characterise low haemoglobin levels as a marker of food insecurity remains in question, given that women were largely recruited near term.”